# Lane-Level Regional Risk Prediction of Mainline at Freeway Diverge Area

**DOI:** 10.3390/ijerph19105867

**Published:** 2022-05-11

**Authors:** Nengchao Lyu, Jiaqiang Wen, Wei Hao

**Affiliations:** 1Intelligent Transportation Systems Research Center, Wuhan University of Technology, Wuhan 430063, China; lnc@whut.edu.cn (N.L.); wenjq@whut.edu.cn (J.W.); 2National Engineering Research Center for Water Transport Safety, Wuhan 430063, China; 3Hunan Key Laboratory of Smart Roadway and Cooperative Vehicle-Infrastructure Systems, Changsha University of Science and Technology, Changsha 410205, China

**Keywords:** regional risk prediction, roadside observation data, surrogate safety measure, feature analysis, catastrophe theory

## Abstract

Real-time regional risk prediction can play a crucial role in preventing traffic accidents. Thus, this study established a lane-level real-time regional risk prediction model. Based on observed data, the least squares-support vector machines (LS-SVM) algorithm was used to identify each lane region of the mainline, and the initial traffic parameters and surrogate safety measures (SSMs) were extracted and aggregated. The negative samples that characterized normal traffic and the positive samples that characterized regional risk were identified. Mutual information (MI) was used to determine the information gain of various feature variables in the samples, and the key feature variables affecting the regional conditions were tested and screened by means of binary logit regression analysis. Upon screening the variables and corresponding labels, the construction and verification of a lane-level regional risk prediction model was completed using the catastrophe theory. The results showed that lane difference is an important parameter to reduce the uncertainty of regional risk, and its odds ratio (OR) was 16.30 at the 95% confidence level. The 10%-quantile modified time to collision (MTTC) inverse, the speed difference between lanes, and 10%-quantile headway (DHW) had an obvious influence on regional status. The model achieved an overall accuracy of 86.50%, predicting 84.78% of regional risks with a false positive rate of 13.37% and 86.63% of normal traffic with a false positive rate of 15.22%. The proposed model can provide a basis for formulating individualized active traffic control strategies for different lanes.

## 1. Introduction

Traffic safety problems are especially compounded in developing countries, yet simple preventive measures can effectively reduce the number of accidents and thus the number of casualties [1]. Therefore, traffic safety management has become a primary preventative measure [2]. Active traffic management systems (ATMS) have been widely used in developed regions, like the United States and European countries, to improve road traffic safety [3]. Different from previous passive control measures, ATMS have the functional characteristics of real-time proactive management and control, taking real-time traffic risk prediction as the core. Based on the predictions, proactive interventions, such as onboard display devices and roadside variable speed limit signs, can be used to warn drivers [4], while rescue vehicles can be readied in advance of when a high-risk situation is predicted [5]—all with the aim to reduce or eliminate collisions. To achieve this, traffic risk prediction models serving ATMS and real-time crash prediction models (RTCPM) have been studied intensively [6].

Real-time risk prediction has been carried out from different angles all around the world. Wang et al. [7] proposed a Bayesian logistic regression model for single-vehicle and multi-vehicle collisions on expressway ramps; then, they explored the impact of real-time traffic variables and weather on accidents. Ma et al. [8] introduced an hourly conflict risk index (HCRI) and used a multiple linear regression model to establish a traffic conflict prediction model for the expressway diversion area and verified the performance of the model with a goodness of fit of 0.945. Based on accident data and traffic operation data from six urban expressways in Shanghai, Peng et al. [4] established a random sampling cost-sensitive multi-layer perceptron (MLP) model and a Rusboost model through comprehensive optimization of output, data, and algorithm levels. The model achieved an average sensitivity and specificity of 81%. Parsa et al. [9] demonstrated the performance of support vector machine (SVM) and probabilistic neural network (PNN) models in real-time accident detection using traffic data at the time of expressway accidents. The overall accuracy of the SVM was higher, but the PNN had a higher rate of accurate accident detection. Li et al. [6] used a long short-term memory convolutional neural network (LSTM-CNN) to establish a real-time collision risk prediction model on arterials to capture time-variant and time-invariant characteristics; the model achieved the greatest sensitivity of 88%.

Data from different sources, specific traffic objects, and diverse modeling methods constitute the bulk of existing risk prediction research, while sufficient relevant traffic data is the foundation for establishing such models. According to the existing literature, the data sources used for traffic risk prediction models are generally divided into two categories. The first type of data comes from the traffic monitoring system [10], which completes the work of risk prediction modeling by extracting traffic flow parameters. Such monitoring systems include dual-loop detectors [11], single-loop detectors [12], and automatic vehicle identification (AVI) [13,14,15]. The second type of data comes from vehicle trajectories [10]; vehicle sensor technology is used to extract vehicle information within a certain range. Traffic data has also been obtained from other sources and used for real-time safety analysis. For example, the risk of urban arteries was studied using Bluetooth data [16]. In terms of data sources, only those that have continuous monitoring and dense distribution can collect reliable enough data for accurate risk prediction [17]. For example, due to mobility and sensing distance limitations, vehicle sensors cannot always accurately extract traffic flow information in the area where they are located. Other types of data also prove difficult to extract the micro-features between regional vehicles. Therefore, although the above methods can provide basic data for traffic risk prediction, there are still shortcomings in satisfying regional macro parameter calculation and micro behavior capture at the same time. However, with the development and progress of intelligent perception technology, fixed roadside sensors can collect refined enough traffic data on road sections, solving the problem of stringent requirements for equipment distribution density and data granularity, which provides great help for real-time risk prediction [18]. For this reason, a fixed roadside sensor was professionally installed and used in this study.

Furthermore, it is critical to select a specific research area for traffic risk prediction modeling. Generally, the research areas are of two types based on their differences in spatial scope. The first type of research area looks at the road in a relatively broad sense; it does not limit its specific attributes. This type of research is not carried out for a specific node, road section, or special alignment, but rather usually refers to a certain category of road, such as freeways [9,19], urban expressways [4,14], or urban arteries [16,20]. The second type of research area looks at the road in a relatively narrow sense; it defines the attributes or structural characteristics of the road. Such research often refers to certain road segments or special nodes, which are part of the overall road, such as ramps [7], diversion areas [8], bridges, tunnels, etc. Most of the existing research focuses on type one—freeways and urban expressways—aiming to achieve road-level risk prediction. A small part of the existing research focuses on type two—key areas, such as diversions and ramps—aiming to achieve segment-level risk prediction. Although there are differences in the spatial scale of the two research areas, they both focus on the relative macro-level traffic risks, while few studies have focused on the regional risk at the lane level. This study aims to bridge this gap, because due to the difference in speed limit and traffic flow stability between the inner and outer lanes of a freeway, different lanes may have different risks. Therefore, this study performed traffic risk prediction at the lane level by dividing the main lanes near the diversion area.

The choice of input and output of a traffic risk prediction model is also critical to its accuracy. Some previous studies mainly used statistical regression methods and data mining algorithms to predict the risk of a freeway [5]. For statistical regression methods, the calculation is convenient and fast due to the fixed mathematical structure. In contrast, data-driven models have strict requirements on data magnitude and data categories due to their learning characteristics, which directly affect the performance of the model. Hence, it has also become a tendency to use non-learning modeling methods to study risk prediction [21]. With modeling, it is extremely important to select relevant candidate variables that describe risk [22]. Environment, traffic, vehicles, and drivers are the main factors involved in risk [23]; the former two factors are more important and easier to collect data for than the latter two. As such, average speed, traffic flow, and occupancy rate and their variation form are widely used as important input variables in prediction models [4,5]. Real-time weather data [9,19,24] is also frequently considered in risk prediction models. In addition, a few studies have incorporated signal data into risk prediction models [6]. In terms of model output, traffic accident data [9,25] is often the best choice for training or validating models. However, due to the difficulty in obtaining accident data, adopting traffic conflicts instead of accident data has become a new alternative [26,27]. From the perspective of variable selection, existing prediction models mainly use traffic road data and weather data as important inputs, but rarely consider the micro-features describing group driving behavior in the research area [18,28]. Therefore, this study used refined traffic data to extract rich traffic parameters and surrogate safety measures (SSMs) and then constructed a regional risk prediction model.

The purpose of this study was to predict regional risk at the lane level. The overall framework is shown in Figure 1. Firstly, on the basis of robust roadside perception data, the least squares-support vector machines (LS-SVM) algorithm was used to identify the main lanes near the diversion area, and various traffic parameters and SSMs were calculated. Secondly, the lane segment was taken as the research area, the traffic conflict in the area was extracted as the risk labels, and the parameters of each area were aggregated as the traffic features within a certain time window. Then, the key feature variables related to regional risk were screened using mutual information (MI) and binary logit regression analysis. Finally, according to the selected variables and corresponding regional status labels, a lane-level regional risk prediction model based on the catastrophe theory was established, evaluated, and verified.

The rest of this paper is organized into sections: Section 2 presents the key methods used in this study, including the LS-SVM algorithm, SSMs, MI, and catastrophe theory. Section 3 describes the experimental data collection and processing, mainly including the definition of regional status and the construction of sample data sets. Section 4 presents the modeling process and results analysis. Section 5 presents the conclusion and future work.

## 2. Methodology

### 2.1. Lane Identification Based on the LS-SVM Algorithm

The LS-SVM algorithm has a faster solution speed and higher convergence accuracy. Therefore, to obtain roadside observation data, the LS-SVM algorithm was used to identify the road center line and edge line to further extract the traffic parameters of the main lane near the diversion area.

The LS-SVM algorithm started from the loss function and used (1) the two-norm in the objective function of its optimization problem and (2) the equality constraints to replace the inequality constraints in the standard SVM algorithm so that the LS-SVM algorithm optimization problem became the solution of linear equations [29].

The LS-SVM model was used in original space y(x)=ωTφ(x)+b,x∈ℝn,y∈ℝ and given the training set {xi,yi}i=1N and the optimization objective:(1)minω,b,eJ(ω,e)=12ωTω+12γ∑k=1Nek2
(2)s.t.                  yk=ωTφ(xk)+b+ek,      k=1,⋯,N

The Lagrange multiplier method was used to transform the original problem into a single parameter, that is, the maximal value problem of α, and finally the regression function was obtained.
(3)$$y\left(x\right)=\sum_{k=1}^{N}\alpha_{k}K\left(x,x_{k}\right)+b$$

The radial basis function (RBF) was used as the kernel function,
(4)$$K\left(x,x_{k}\right)=\exp\left[-\frac{{\left(x-x_{k}\right)}^{2}}{2\sigma^{2}}\right]$$
where, the error ***e*** is a slack variable, and its significance is to introduce outliers in the support vector; and γ is the regularization parameter, which determines the trade-off between training error minimization and smoothness. In the common case of a Gaussian RBF kernel, $\sigma^{2}$ is the square of the bandwidth, α is the Lagrange multiplier, which is also a support value, and *b* is a real constant.

### 2.2. Extraction of SSMs

SSMs are a series of parameters developed to describe traffic conflicts based on time, distance, or deceleration. SSMs are often used to measure the microscopic behavior between vehicles, which can further characterize the traffic safety level of the area. Therefore, based on the collected observation data, relevant SSMs were selected to obtain the traffic state information of the research area. In addition to standard indicators, such as time to collision (TTC), time headway (THW), and distance headway (DHW), the following SSMs were taken into consideration.

(1) Modified Time to Collision [30]

Modified time to collision (MTTC) uses the accelerations of the preceding and following vehicles to estimate the TTC of a car-following event, without assuming constant speeds of the preceding and following vehicles. Given the inherent differences between not just each vehicle but also each driver on the road, MTTC considers various dangerous situations in the longitudinal car-following scenario, and its calculation method is as follows:(5)$$\mathrm{MTTC}=\frac{-\Delta V\pm \sqrt{\Delta V^{2}+2\Delta aD}}{\Delta a}$$
where, ΔV represents relative speed; Δa represents relative acceleration; *D* represents initial relative distance.

(2) Potential Index for Collision with Urgent Deceleration (PICUD) [31]

PICUD represents the distance between two vehicles when they come to a complete stop with emergency braking. PICUD is suitable for evaluating the collision risk of consecutive vehicles with similar speeds. The calculation method is as follows:(6)$$\mathrm{PICUD}=\frac{V_{l}^{2}-V_{f}^{2}}{2a_{s}}+D-V_{f}\Delta t$$
where, $V_{f}$ and $V_{l}$ represent the speed of the following vehicle and the preceding vehicle, respectively, $a_{s}$ represents the emergency deceleration rate when the vehicle stops, and Δt represents the driver’s reaction time.

(3) Deceleration Rate to Avoid a Crash (DRAC) [32]

DRAC refers to the minimum deceleration rate required by a vehicle to avoid a collision with another vehicle. It takes into account the effect of different speeds and decelerations in the flow of traffic and is calculated as follows:(7)$$\mathrm{DRAC}=\frac{{\left(V_{f}-V_{l}\right)}^{2}}{2\left(P_{l}-P_{f}-L\right)}$$
where, $P_{l}$ and $P_{f}$ represent the positions of the preceding and following vehicles, respectively, and *L* represents the length of the preceding vehicle.

(4) Difference of Space Distance and Stopping Distance (DSS) [33]

DSS is defined as the difference between the space distance and the stopping distance, taking into account the influence of road conditions on vehicle motion. If the driving distance of the following vehicle exceeds the sum of the driving distance of the preceding vehicle and the initial relative distance, that is, the DSS is a non-positive value, then this creates a collision risk. The calculation method is as follows:(8)$$\mathrm{DSS}=\left(\frac{V_{l}^{2}}{2\mu_{fric}g}+D\right)-\left(V_{f}\Delta t+\frac{V_{f}^{2}}{2\mu_{fric}g}\right)$$
where, $\mu_{fric}$ is the friction coefficient, and g is the acceleration of gravity.

### 2.3. Selection of Related Variables Based on MI

While there are many variables that can characterize regional risk, it is extremely important to select appropriate feature variables for modeling. Using too many feature variables will inevitably increase the model’s calculation load and solution time; therefore, because some features have little contribution to the results, they can and should be ignored in modeling.

In probability theory and information theory, the mutual information or trans-information of two random variables is a measure of the interdependence between variables. In other words, because one random variable is known, the uncertainty of another random variable is reduced accordingly. MI describes the mutual dependence between two event sets, which can be used to capture any relationship (including linear and nonlinear relationships) between each feature and label and return the estimation of MI. The MI of two discrete random variables *X* and *Y* can be defined as:(9)$$I\left(X;Y\right)=H\left(X\right)-H\left(Y|X\right)=\sum_{y\in Y}\sum_{x\in X}p\left(x,y\right)\log\left(\frac{p\left(x,y\right)}{p\left(x\right)p\left(y\right)}\right)$$
where, H(X) and H(Y) are the information entropy of *X* and *Y*, respectively. H(Y|X) represents the information entropy of random variable *Y* under the premise of known random variable *X*. p(x,y) is the joint probability distribution function of *X* and *Y*. p(x) and p(y) are the marginal probability distribution functions of *X* and *Y*, respectively. Mutual information I(X;Y) is the relative entropy of joint distribution p(x,y) and marginal distribution p(x)p(y), in bits.

### 2.4. Regional Risk Prediction Modeling Based on Catastrophe Theory

The evolution of regional risk describes a process in which the safety level of the research area transitions from a steady state to an unsteady state or vice versa. The catastrophe theory is often used to study such discontinuous changes in systems [34], and it is based on mathematical tools, such as the singularity theory and topology.

In this study, to describe the change process of the regional risk state, the regional safety level was defined as the state variable, and the elements affecting the regional safety level were defined as the control variable. The change of state variables over time can be expressed by the balance surface function:(10)$$\frac{\partial s}{\partial t}=\frac{\partial F\left(s;\mu ,\nu ,\lambda \right)}{\partial s}$$
where, F(s;μ,ν,λ) is the potential function of the catastrophe model, *x* is the state variable, and μ,ν,λ is the control variable. Each control variable was measured by the feature elements it contained:(11)$$\left[\mu \;\;\;\;\;\nu \;\;\;\;\;\lambda \right]=EA=\left[\begin{array}{cccc} e_{1} & & & \\ & e_{2} & & \\ & & \ddots & \\ & & & e_{n} \end{array}\right]\left[\begin{array}{ccc} a_{1} & b_{1} & c_{1}\\ a_{2} & b_{2} & c_{2}\\ \vdots & \vdots & \vdots \\ a_{n} & b_{n} & c_{n} \end{array}\right]$$
where, **E** represents the feature sequence that is correlated with the regional risk status, and **A** represents the composition matrix of feature elements under each type of control variable.

Control variables and their feature elements were implemented in the catastrophe framework. To facilitate the calculation, each element in the three types of control variables μ,ν,λ was standardized.

The benefit and cost elements were treated as follows, respectively:(12)$${\hat{z}}_{ij}=\frac{z_{ij}-z_{\min}}{z_{\max}-z_{\min}},\;{\hat{z}}_{ij}=\frac{z_{\max}-z_{ij}}{z_{\max}-z_{\min}}$$
where, $z_{ij}$ represents the value of the *j*-th feature of the *i*-th type of control variable.

After completing the dimensional unification of each feature element, the normalization formula of the catastrophe model was used to obtain the catastrophe value of the control variable:(13)$$d_{i}=\frac{1}{h_{i}}\sum_{j=1}^{h_{i}}{\left(\frac{{\hat{z}}_{ij}}{\alpha_{jh_{i}}}\right)}^{{\left(j+1\right)}^{-1}}$$
where, $\alpha_{jk_{i}}$ represents the coefficient corresponding to the *j*-th feature element in the case of containing $h_{i}$ control variables, that is, the coefficient corresponding to the control variable in the bifurcation point set equation, which was determined by the potential function *F*. $d_{i}$ represents the catastrophe value of a certain type of control variable μ,ν,λ.
(14)$$R_{s}=\{\begin{array}{c} 1,\;\;\;\;\;\;\;\;\;\min\left\{d_{1},d_{2},\ldots ,d_{I}\right\}\leq R_{80}\\ 0,\;\;\;\;\;\;\;\;\;\min\left\{d_{1},d_{2},\ldots ,d_{I}\right\}>R_{80} \end{array}$$
where, $R_{s}$ represents the regional risk status value, $d_{I}$ represents the catastrophe value of the class *I* control variable, and $R_{80}$ represents the 80th percentile value of the safety level.

## 3. Data Collection and Processing

### 3.1. Roadside Observation Experiment

To obtain the data set required for risk prediction, a roadside observation experiment was carried out on a freeway in Guangdong Province in China. Considering the need for rapid extraction of conflict events in this study, a diversion area of the freeway was selected as the observation point. By installing a fixed microwave detector on a roadside vertical pole, information was collected on moving vehicles in that particular area of the road. The observation time was from 15–17 October 2020 (14:42 on Thursday to 18:41 on Saturday), and continuous traffic data in the observation area was obtained for 52 h. The time range of the experiment included weekdays and weekends, which made the data more comprehensive and representative and played a positive role in verifying the universality and stability of the proposed risk prediction model. The observation area included a mainline and an off-ramp. The mainline comprises a two-way four-lane section, and the off-ramp comprises two lanes. The observation location and observation area are shown in Figure 2.

The roadside observation data collection system was composed of a wide-area microwave detector, a micro industrial computer, and a power supply, as shown in Figure 3. The wide-area microwave detector supported multi-target trajectory tracking detection in multiple lanes, and the longitudinal view reached 300 m (model: DTAM D29; item: wide-area radar microwave vehicle detector; manufacturer: Nanjing Huiershi Intelligent Technology Co., Ltd., Nanjing, China). In the experiment, the detector was set to output basic information, such as longitudinal position, lateral position, longitudinal speed, lateral speed, and the ID and time stamp of the vehicles in the observation area at a frequency of 1–20 Hz. The micro industrial computer used the network transmission protocol and built-in program to receive the sensing data of the microwave detector in real time and store it in the form of text. The power supply powered the microwave detector and the micro industrial computer at the same time and output 12 V DC voltage. The experiment was approved by the expressway management, and a professional construction team installed equipment.

### 3.2. Data Processing

#### 3.2.1. Raw Data Processing

Taking the position of the microwave detector as the base point, the raw trace file included the traffic flow data in both directions towards and away from the base point. In the observation coordinate system of the microwave detector, the longitudinal speed of the vehicle far from the detector was defined as positive, while that of the vehicle close to the detector was defined as negative. Since this study only focused on the traffic safety conditions in the diversion area (near the off-ramp), the longitudinal speed of the point trace was used to distinguish traffic flow in two directions, and all traffic data from vehicles leaving the base point was excluded. The scattered point traces were converted into continuous trajectories through the processing program, and the processing flow is shown in Figure 4.

The processing of point traces mainly included three processes. Firstly, according to the ID assigned to each vehicle by the microwave detector, the point traces of the same vehicle were associated, and the initial trajectory file was obtained. Secondly, it was necessary to check whether each trajectory was missing, and the nearest neighbor idea and interpolation method were used to supplement any missing points to obtain the intermediate trajectory file. Finally, if the same vehicle led to two or more short trajectories due to the misassignment of IDs, then the information of the next moment was predicted by the end point of the trajectory and compared with the information of the start point of the new trajectory generated at the same moment, so as to complete the splicing of the short trajectory belonging to the same vehicle. Based on the described processing steps, the complete trajectory of each passing vehicle in the research area was obtained.

#### 3.2.2. Lane Identification in the Research Area

The trajectory file described above retained the traffic flow information heading to the microwave detector (single direction), but the position of each vehicle was defined relative to the microwave detector. This cannot express the distribution of vehicle trajectories in different lane areas, and it is also inconvenient to extract traffic parameters in each lane area. Therefore, based on the location information of the trajectory data, the LS-SVM algorithm was used to complete the lane identification of the mainline and the ramp in the diversion area, as shown in Figure 5, where the red dotted line represents the road center line, and the green solid line represents the road edge line. The origin of the coordinates represents the position of the microwave detector; it should also be noted that the driving direction of the vehicles was from right to left, as they approached the microwave detector. The horizontal axis represents the longitudinal distance of the point trace relative to the microwave detector, and the vertical axis represents the lateral distance of the point trace relative to the microwave detector. It can be observed from Figure 5 that due to the installation height, the range of 0–20 m away from the microwave detector could not be captured. As such, the monitoring length of the microwave detector to the main line was about 230 m, and the number of passing vehicles captured was about 48,000.

While using the LS-SVM algorithm to identify the lanes in the research area, the expression function of the road centerline was also obtained (i.e., Equation (3)). In real time, the expression function was used to determine the lane where the trajectory point was located. Due to the difference in speed limits on different lanes of the mainline and the traffic disturbance caused by vehicles leaving, and due to the fact that the traffic movement of the off-ramp was relatively stable, this study only analyzed the data of the mainline.

### 3.3. Dataset Construction

#### 3.3.1. Indicator Calculation and Conflict Event Extraction

According to the trajectory point information of all vehicles in the same lane at the same time, by establishing adjacent vehicle pairs, various parameters related to driving safety were extracted, including position, speed, acceleration, lane number, TTC, THW, DHW, MTTC, PICUD, DRAC, DSS, and other indicators.

After the lane area was determined, the conflict event was extracted by the TTC and deceleration threshold to characterize the risk level of the lane at a certain moment, and it was used as a sample label for subsequent modeling. Based on existing literature [35,36], the TTC threshold of 3 s and the severe deceleration threshold of −2.943 m/s^2^ were selected to determine conflict events, and a total of 159 conflict events were extracted. 83 conflict events were extracted in the outer lane area of the mainline, and 76 conflict events were extracted in the inner lane area of the mainline.

The distributions of the conflict events and non-conflict events on longitudinal speed were compared for the outer lane and the inner lane, as shown in Figure 6 and Figure 7, respectively. It can be observed from Figure 6 that on the outer lane, the average speed of conflict events (82.2 km/h) was lower than that of non-conflict events (87.5 km/h), and the t-test (two-sample heteroskedasticity hypothesis) results show that there was a significant difference in speeds between the two groups (*p* = 0.03 < 0.05). Similarly, it can be observed from Figure 7 that on the inner lane, the average speed of conflict events (89.7 km/h) was lower than that of non-conflict events (100.4 km/h), and the result of the t-test showed that there was a significant difference in speed between the two groups (*p* = 0.002 < 0.05). It indicated that lower speed may lead to higher collision risk [37], which was consistent with the conclusion of literature [8] that speed is negatively correlated with traffic conflict risk. Furthermore, the differences in the speed distribution of the conflict events in the outer lane and the inner lane were also compared, as shown in Figure 8. The average speed of conflict events in the outer lane (82.2 km/h) was lower than that in the inner lane (89.7 km/h), but the t-test showed that the difference in speed between the two groups was not significant (*p* = 0.07 > 0.05).

#### 3.3.2. Sample Construction of Lane-Level Area

Constructing a dataset with positive and negative samples is a prerequisite for regional risk prediction modeling. Positive samples refer to samples with conflict events in the area, and negative samples refer to samples with normal traffic conditions in the area. In this study, the samples were composed of feature variables and labels. The regional status (whether or not conflict occurred) was used as the label, and the traffic parameters within a certain time window before the regional status occurred were extracted as feature variables. It has been found in previous studies that the closer the traffic flow data is captured to the time when the regional situation occurs, the more beneficial it is to the risk prediction model [38]. Therefore, taking the occurrence time of regional status as the critical point, the segment 30–60 s before the critical point was selected as the time window [18]. The traffic parameters within 30 s were counted as the feature variables corresponding to the regional status, so as to complete the construction of each sample. The segment 0–30 s before the critical point was reserved as the response stage of risk control.

Since the feature variables covered a variety of traffic parameters within 30 s, it was necessary to aggregate indicators to characterize the traffic conditions in the area. Specifically, the mean value of a parameter of the area in every 1 s was calculated, and then the differentiated feature variables within 30 s were obtained by extracting the mean, standard deviation, range, and quantile value. Finally, three feature sets of regional macro-traffic parameters, regional micro-SSMs, and inter-regional parameter differences were formed, with a total of 51 feature variables, as shown in Table 1. The SSMs in the table refer specifically to 1/TTC, 1/MTTC, DRAC, PICUD, and DSS.

The sample construction followed these steps:(1)Taking the conflict events extracted from each lane area as labels, the regional feature variables corresponding to the labels were extracted to complete the construction of positive samples.(2)On the continuous time series, the time range covered by the positive samples in each region was excluded, and the construction of the negative samples was completed. At the same time, by checking the samples, the samples with missing features and abnormal eigenvalues were eliminated.(3)According to the method, 153 positive samples representing conflicts and 2019 negative samples representing normal traffic were established for the area.

## 4. Results and Discussion

### 4.1. Information Gain of Feature Variables

The calculated feature variables were parameters that directly or indirectly affected the model results and were used as inputs to the model. However, some feature variables were strongly correlated with labels (i.e., model output), while some feature variables were weakly correlated with labels. To ensure accuracy, the time and space complexity of the model was reduced by decreasing the weakly correlated feature variables. Therefore, similar important variable ranking methods in existing studies were followed [5,6,19]. In other words, this study constructed a regional risk prediction model by screening out some feature variables to increase prediction accuracy. In the data processing, 51 feature variables belonging to three types of feature sets were extracted. MI was used to capture the relationship between each feature variable and label in the sample, and the information gained from each feature variable was obtained. According to the reduction degree of the information gain to the uncertainty of the results, the information gains of the 51 feature variables were arranged in descending order. In consideration of space, just the top 10 feature information gains are shown in Figure 9. The top 10 most important features for regional risk prediction were: lane number, 1/MTTC 10% quantile, speed difference between lanes, DHW 10% quantile, THW 5% quantile, THW 10% quantile, DHW 5% quantile, 1/MTTC mean, speed mean, and speed range.

The same indicator produced different variables due to the expression form, such as THW 5% quantile and THW 10% quantile, DHW 10% quantile, and DHW 5% quantile, which had the same effect on the results. Therefore, in consideration of complexity, the multi-collinearity problem was avoided by discarding some variables [14], and different types of variables were screened for modeling. Among all the feature variables, the lane number had the largest information gain, which indicated that the regional risk between the outer lane and inner lane of the mainline at the diversion area was different, which is in line with the original assumption of this study. Second, the 1/MTTC 10% quantile was used as an SSM to describe the microscopic behavior of vehicles, which helped to explain the regional risk caused by car-following events. The speed difference between lanes showed that if the speed difference between the outer and inner lanes was too large, then regional conflict in the lanes may be induced. Previous studies have also shown that vehicle speed is an important factor affecting traffic risk [9,19]. The DHW 10% quantile and the THW 5% quantile indicated that close car-following had a positive effect on in-lane conflict events. In Figure 9, it can be observed that the low percentile of micro behavior indicators accounted for 50% of the top 10 feature variables, which showed that the low percentile better characterized the risk tendency of the area than the mean, which is consistent with existing research conclusions [36].

### 4.2. Impact Analysis of Important Variables on Regional Risk

Although MI screened some feature variables that reduce the uncertainty of the regional risk level, the effect of each feature variable on the regional risk was still unclear. Therefore, to improve the overall performance, binary logit regression analysis was used to further explore the influence of screening feature variables on regional status. According to the value characteristics of the label, the regional status (classified data) was used as the dependent variable. According to the meaning and value range of feature variables, lane number (classified data) was used as the classification covariate, while 1/MTTC 10% quantile, the speed difference between lanes, DHW 10% quantile, and THW 5% quantile (quantitative data) were used as covariates. After establishing the binary logit regression test model, the analysis was completed, the results of which are shown in Table 2. The result of the Likelihood Ratio Test showed that the test model was effective (*p* = 0.000 < 0.05). The result of the Hosmer and Lemeshow Test indicated that the information in the data had been fully extracted (*p* = 0.672 > 0.05), and the model had a high goodness of fit.

To further understand the relative trend of each feature variable, Figure 10 presents the odds ratio (OR) at the 95% confidence level. It can be observed from Figure 10 that the lane number showed a significant level of 0.01 (*p* = 0.000 < 0.01), which indicated that the lane number had a significant positive impact on the regional status. Different lanes have different speed limits, which may generate different degrees of collision risk [39]. There are also studies that believe that drivers need to devote more attention and make more efforts under different speed limit conditions in the inner and outer lanes, and the traffic collision risk and safety level may be affected [40]. In addition, the OR of the lane number indicated that each time the lane changed by one unit, the regional status changed by a factor of 16.30. Similarly, the speed difference between lanes (*p* = 0.045 < 0.05) and 1/MTTC 10% quantile (*p* = 0.043 < 0.05) both showed significance at the 0.05 level. When the speed difference between lanes changed by one unit, the change range of the regional status was 0.97 times. When the 1/MTTC 10% quantile changed by one unit, the change range of the regional status was 1.02 times. The DHW 10% quantile (*p* = 0.001 < 0.01) showed significance at the 0.01 level, and each time it changed by one unit, the regional status changed by a factor of 0.99. Taken together, the ORs for these feature variables suggest that they affected the regional status. However, the THW 5% quantile did not show significance (*p* = 0.373 > 0.05), suggesting that it had a weak effect on the regional status.

Based on this analysis and after synthesizing the information gain and influence direction of feature variables on regional status, four feature variables were selected—lane number, 1/MTTC 10% quantile, the speed difference between lanes, and DHW 10% quantile—as the inputs of the proposed regional risk prediction model.

### 4.3. Regional Risk Prediction Modeling

Based on the positive samples and negative samples in the aforementioned datasets, the training and test sets were allocated in a ratio of 7:3. The training set included 107 positive samples representing risk areas and 1413 negative samples representing normal areas, while the test set included 46 positive samples and 606 negative samples. Combined with the catastrophe theory and statistical feature variables, the control variables of the catastrophe model were divided into three categories: regional macro-traffic parameters, regional micro-SSMs, and inter-regional parameter differences, and the state variable was the regional status. The four most critical feature variables were identified by MI and feature influence analysis. The regional macro-traffic parameter used for modeling was the lane number, while the regional micro SSMs used for modeling were the 1/MTTC 10% quantile and DHW 10% quantile, and the inter-regional parameter difference used for modeling was the speed difference between lanes. Considering that the catastrophe model is a non-learning model, the output results (predicted labels) of the model were compared with the corresponding true labels on the test set. The prediction results of the regional risk prediction model based on the catastrophe theory are shown in Table 3. The normal area was regarded as one type of result, the risk area was regarded as another type of result, and the regional risk prediction results on the inner and outer lanes were further divided. The accuracy rate of the model was 86.50%, while the true positive rate (TPR) and false positive rate (FPR) of the risk area were 84.78% and 13.37%, respectively, and the TPR and FPR of the normal area were 86.63% and 15.22%, respectively.

Table 3 shows the performance of the model in risk prediction on the outer lane and inner lane. For the outer lane, the accuracy rate of regional risk prediction was 87.50%; for the inner lane, the accuracy rate of regional risk prediction was 66.67%. Comparing the traffic flow in the outer lane and the inner lane, the diversion area experienced a greater traffic speed disturbance in the outer lane, making the traffic fluctuations in the outer lane more frequent, so the model was more sensitive to the impending dangerous situation. However, the design speed of the inner lane was higher than that of the outer lane, and vehicles drove at a relatively uniform speed and were not as affected by the traffic flow in the other lanes in the same direction. The traffic flow was relatively stable in the inner lane, so the model had a slightly poorer effect on risk identification there.

To compare the performance differences between the proposed model and other models, regional risk prediction based on naive Bayes (NB), K-nearest neighbor (KNN), support vector machines (SVM), and decision tree (DT) models were carried out. Considering that the large difference in positive and negative sample sizes in the training set may have led to the overfitting of such machine learning models, the synthetic minority oversampling technique (SMOTE) was used to solve the problem of sample imbalance. The processed training set included 460 positive samples and 1688 negative samples, and the test set used to verify the effect of the model was not processed. The prediction results of each model for lane risk are shown in Table 4.

According to the prediction results, the catastrophe theory model predicted 84.78% of the risk areas with a false positive rate of 13.37%; the SVM model predicted 86.96% of the risk areas with a false positive rate of 25.41%; the DT model predicted 76.09% of the risk areas with a false positive rate of 16.17%; the KNN model predicted 73.91% of the risk areas with a false positive rate of 18.15%; the NB model predicted 67.39% of the risk areas with a false positive rate of 21.12%. In the prediction of the risk areas, the recall of the SVM model and the catastrophe theory was relatively close and significantly higher than the other models, as shown in Figure 11. In the prediction of the normal areas, the catastrophe theory, DT model, and KNN model outperformed the NB and other models. In terms of overall prediction accuracy, the catastrophe theory, the DT models, and the KNN outperformed the other models. On the whole, the catastrophe model had a better identification ability for both normal areas and risk areas than the other models. The overall accuracy of the catastrophe model was high with a good balance between recall and FPR. In addition, the performance of the learning models may have been limited to a certain extent due to the size of the training dataset and the difference in sample categories. Therefore, the regional risk prediction model based on the catastrophe theory (a non-learning model) outperformed the other learning models.

Summarizing the existing literature and comparing the differences in risk prediction and normal traffic prediction in different studies, the model performance of this study was further evaluated. The results are shown in Table 5. Sensitivity was used to describe the prediction effect of the model for the real-time risk, and specificity was used to describe the prediction effect of the model for normal traffic. It can be observed from Table 5 that compared with the risk prediction effect of the models in other literatures [4,5,14,19], the sensitivity and specificity indicators of the proposed model were improved. While this improvement effect may not have been overly significant, it did indirectly show that the use of lane-level macro-traffic parameters and micro-behavior parameters in this study was of positive significance for predicting regional risks.

## 5. Conclusions

This study took the mainline area of the freeway connecting the off-ramp as the research object. The macro and micro parameters of the outer and inner lanes of the mainline were extracted using roadside observation data, and a sample set of risk areas and normal areas was constructed. Mutual information and binary logit regression analysis were used to determine the key feature variables affecting regional status, and combined with the catastrophe theory, a lane-level regional risk prediction model was established. Finally, the performance of the regional risk prediction was compared by integrating different modeling methods and similar studies. The results showed that:

(1) For the 51 categories of feature variables extracted, mutual information helped determine the information gain of feature variables that reduce regional risk uncertainty. The top 5 feature variables of the information gain were lane number, 1/MTTC 10% quantile, the speed difference between lanes, DHW 10% quantile, and THW 5% quantile.

(2) The binary logit regression analysis showed that the inner and outer lanes had a significant impact on the regional risk level, and the odds ratio of lane difference was as high as 16.30. The 1/MTTC 10% quantile, the speed difference between lanes, and DHW 10% quantile showed different levels of significance, which had an impact on the regional risk. The THW 5% quantile did not show significance, and the effect on the regional risk level was very weak.

(3) The lane-level regional risk prediction model predicted whether there would be a conflict in the area 30 s in advance. The model achieved an overall accuracy of 86.50%, predicting 84.78% of regional risks with a false positive rate of 13.37% and predicting 86.63% of normal traffic with a false positive rate of 15.22%. Compared with the NB, KNN, SVM, and DT models, the proposed model had a better identification ability for normal areas and risk areas. Compared with similar real-time risk prediction studies, the proposed model had a good balance in sensitivity and specificity, showing relatively stable prediction performance.

In this study, a lane-level regional risk prediction model was established. By predicting the risk levels in different lanes of the mainline in real time, it can provide a theoretical basis for research on lane-level active risk management and control strategies. However, due to the limited characteristics of the research area, it is necessary to further verify and debug the robustness and universality of the model in more traffic scenarios in the future. Furthermore, although more macroscopic and microscopic parameters characterizing regional status were extracted in this study, only a few key features were retained for modeling. Therefore, the accuracy of the model can be improved by expanding feature variable extraction and optimizing feature screening methods.

## Figures and Tables

**Figure 1 ijerph-19-05867-f001:**
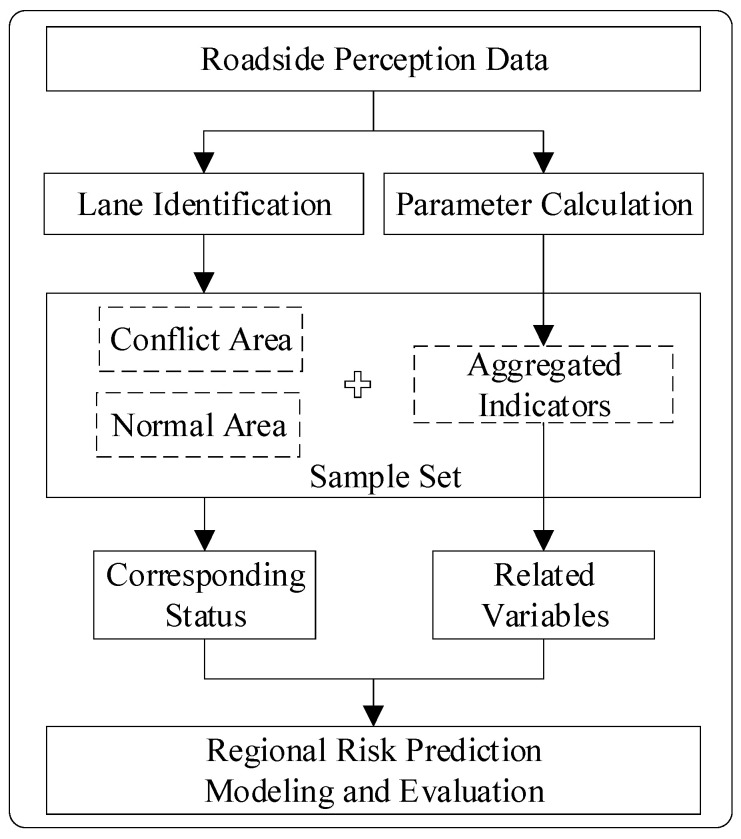
The overall framework of this study.

**Figure 2 ijerph-19-05867-f002:**
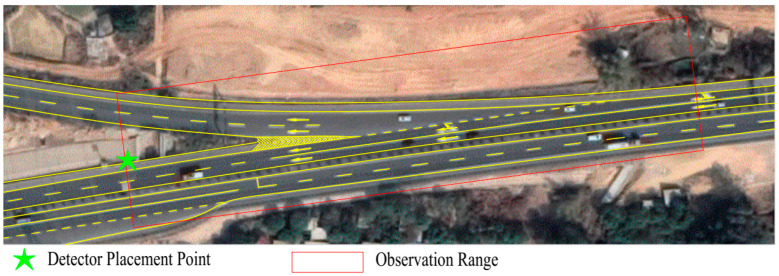
The installation point and observation area of the microwave detector.

**Figure 3 ijerph-19-05867-f003:**
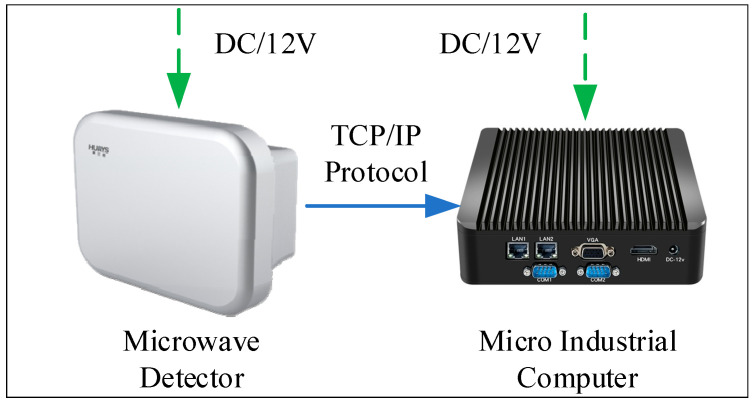
Roadside data collection system.

**Figure 4 ijerph-19-05867-f004:**
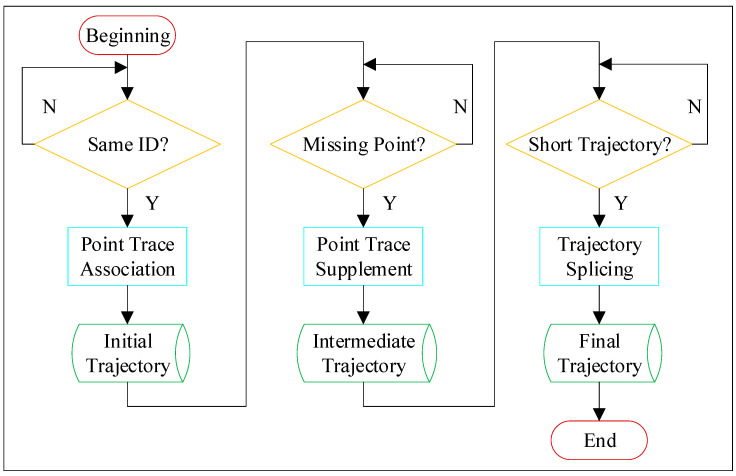
The processing flow of point traces.

**Figure 5 ijerph-19-05867-f005:**
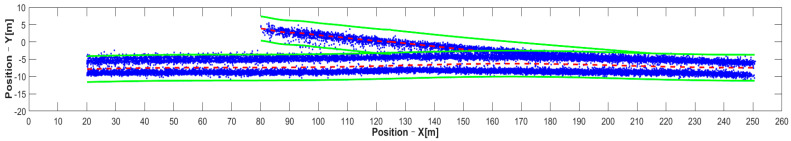
Lane identification in research area.

**Figure 6 ijerph-19-05867-f006:**
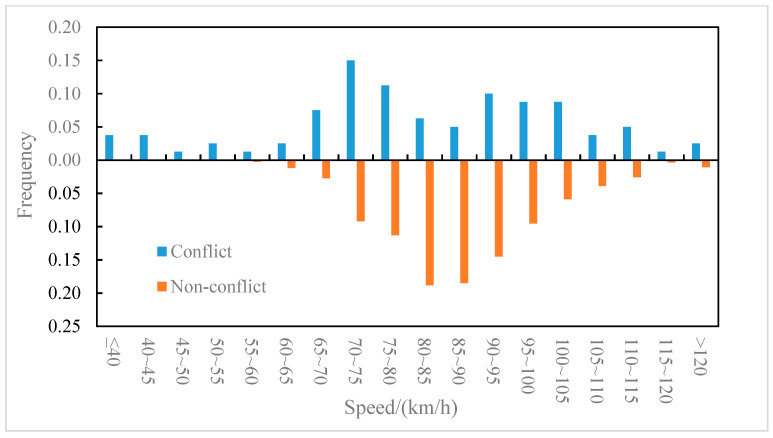
Outer lane—speed distribution of conflict events and non-conflict events.

**Figure 7 ijerph-19-05867-f007:**
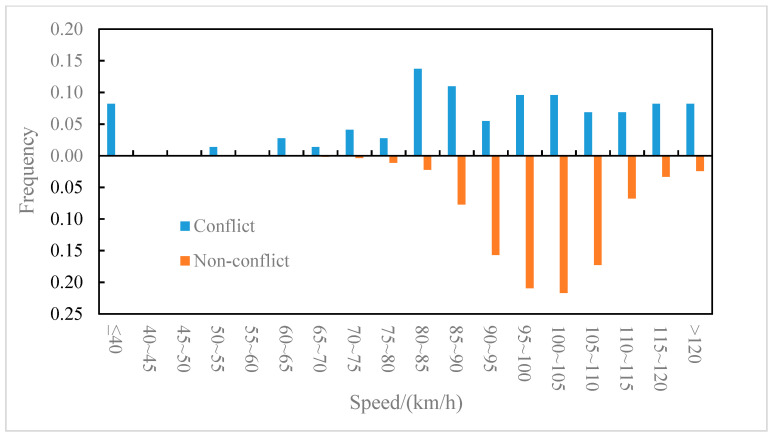
Inner lane–speed distribution of conflict events and non-conflict events.

**Figure 8 ijerph-19-05867-f008:**
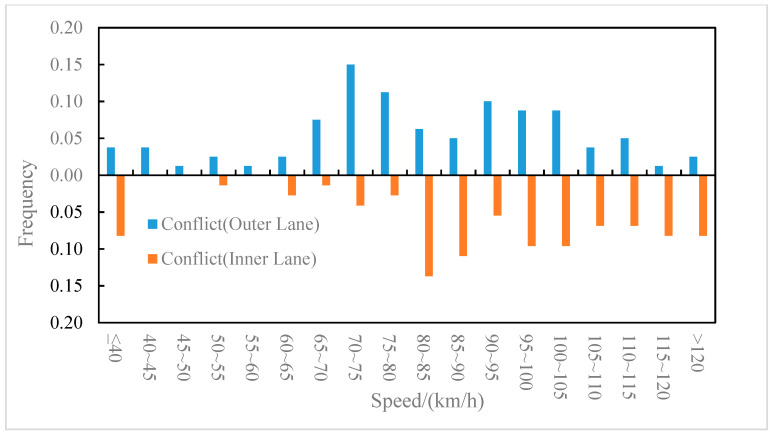
Speed distribution of conflict events on the outer and inner lanes.

**Figure 9 ijerph-19-05867-f009:**
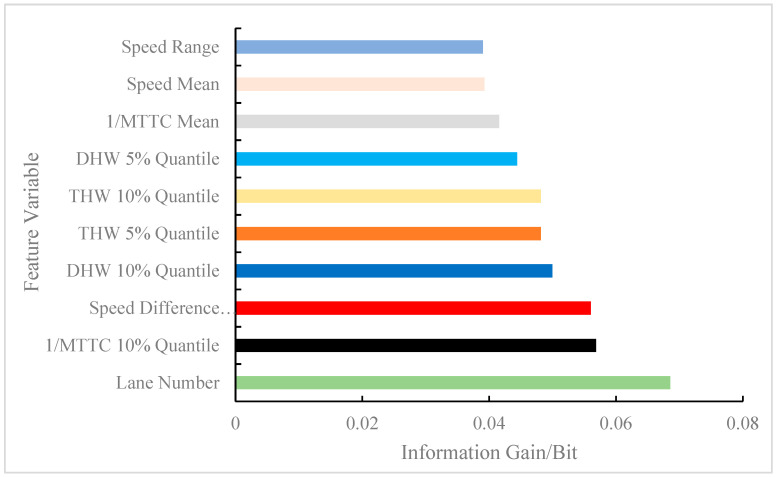
Information gain of feature variables (only the top 10 were counted).

**Figure 10 ijerph-19-05867-f010:**
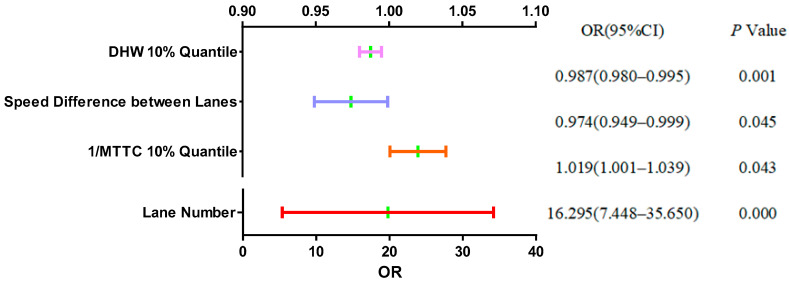
ORs of feature variables.

**Figure 11 ijerph-19-05867-f011:**
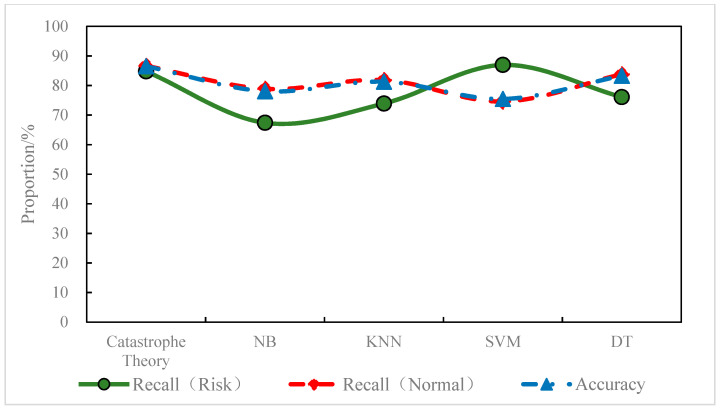
Prediction accuracy of different models for regional status.

**Table 1 ijerph-19-05867-t001:** Feature variables and their meanings.

Category	Variable Name *
Regional Macro-Traffic Parameters	Flow mean/standard deviation/range
	Speed mean/standard deviation/range
	Occupancy mean/standard deviation/range
	Acceleration mean/standard deviation/range
	Lane number
Regional Micro-SSMs	THW mean/standard deviation/10% quantile/5% quantile
	DHW mean/standard deviation/10% quantile/5% quantile
	SSMs mean/standard deviation/10% quantile/5% quantile
Inter-Regional Parameter Differences	THW/DHW difference between lanes
	SSMs difference between lanes
	Flow/speed/occupancy difference between lanes

Variable name *: All variables were statistically obtained based on the average value of a parameter within the 30 s time window of the area.

**Table 2 ijerph-19-05867-t002:** Impact Analysis of feature variables on regional status.

Term	B	S.E.	Wald	Sig.	Exp(B)	95% CI for Exp(B)	
						Lower	Upper
Lane Number	2.791	0.399	48.815	0.000	16.295	7.448	35.650
1/MTTC 10% Quantile	0.019	0.010	4.079	0.043	1.019	1.001	1.039
Speed Difference between Lanes	−0.026	0.013	4.008	0.045	0.974	0.949	0.999
DHW 10% Quantile	−0.013	0.004	11.741	0.001	0.987	0.980	0.995
THW 5% Quantile	0.059	0.066	0.794	0.373	1.061	0.932	1.207
Intercept	−2.826	0.459	37.856	0.000	0.059	-	-

**Table 3 ijerph-19-05867-t003:** Results of regional risk prediction model.

Prediction Results		True Labels		
		Risk Area (Outer Lane)	Risk Area (Inner Lane)	Normal Area
Prediction Labels	Risk Area (Outer Lane)	35	-	81
	Risk Area (Inner Lane)	-	4	0
	Normal Area	5	2	525

**Table 4 ijerph-19-05867-t004:** Regional risk prediction results of different models.

Model	Risk Area		Normal Area		Accuracy
	TPR	FPR	TPR	FPR	
Proposed Catastrophe Theory	84.78%	13.37%	86.63%	15.22%	86.50%
NB	67.39%	21.12%	78.88%	32.61%	78.07%
KNN	73.91%	18.15%	81.85%	26.09%	81.29%
SVM	86.96%	25.41%	74.59%	13.04%	75.46%
DT	76.09%	16.17%	83.83%	23.91%	83.28%

**Table 5 ijerph-19-05867-t005:** Comparison of risk prediction effects in different studies.

Author	Sensitivity	Specificity	Accuracy
Peng et al., 2020 [4]	84.21%	81.62%	-
Basso et al., 2018 [14]	75. 03%	77.53%	-
You et al., 2017 [19]	87.52%	73.22%	80.29%
Sun et al., 2016 [5]	77.90%	79.30%	79.20%
This Study	84.78%	86.63%	86.50%

## Data Availability

The data presented in this study are available on request from the first author. The data are not publicly available due to research team’s dataset management.

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
