# Peer review of "Lane-Level Regional Risk Prediction of Mainline at Freeway Diverge Area"

_ijerph, 2022, doi:10.3390/ijerph19105867_

Round 1

Reviewer 1 Report

The title of this paper is “Lane-level Regional Risk Prediction of Mainline at Freeway Diverge Area.” The manuscript is well organized, the experimental design is sound, and the results are convincing. I only have one major concern and two reminders.

First, the data collected from the Roadside Observation Experiment is the key that affects the verification of the proposed model in this study. The authors conducted this experiment from October 15 to 17, 2020, from Thursday to Saturday. It included two workdays and a weekend, which means the mixing driving pattern may be present in the collected data pool. It would be better to state any observations regarding this issue during the experimental dates and analyze their influence on the model prediction.

Second, the authors used a wide-area microwave detector to collect data. Please provide the product information such as model no., brand name, manufacturer, and country.

Line 313, the 2.943m/s2 should be 2.943m/s^2.

Author Response

Reply, please refer to the attachment

Reviewer 2 Report

A developed paper. It is well researched and put together. The methodology is sound and well demonstrated. A number of improvements are required as per below:

  • Can the introduction section be improved by deliberating more on the aim of the research?
  • Improve the quality of Figure 2. It is difficult to see and observe.
  • For figure 3, provide a more detailed explanation.
  • Figure 11, Prediction accuracy of different models for regional status, is paramount to this research. Thus should be discussed more comprehensively. For example, the prediction effect of risk areas needs to be elaborated on step-by-step.  Discuss the error or margin also and thus the accuracy of this prediction model. Finally, the paper needs to be proofread, as there are some grammar and editing errors that require careful attention.

Reviewer 3 Report

This paper focuses on lane-level real-time regional risk prediction with real traffic data. The topic looks interesting. The paper is well organized. However, I have some comments for the authors to address as follows.

  1. What does “1/modified time to collision (MTTC)” mean in the abstract?
  2. In Equation 5, how do you deal with situations when the relative acceleration is 0?
  3. I think Equation 7 is incorrect. Should it be DRAC = (Vf^2 – Vl^2)/2/(Pl-Pf-L)?
  4. Need to provide more background information about Equation 8. Does the leading vehicle stop suddenly?
  5. In Line 418, “all the feature variables considered are significant”. Since that p=0.373>0.05 for THW 5% quantile, you can’t conclude that.
  6. Provide meanings of all output terms in Table 2, such as B, S.E., Wald, …

Reviewer 4 Report

The authors report an experimental study about real-time risk prediction.

Notes:

The abstract is approximately 290 words, instructions for Authors suggest that the abstract should be a total of about 200 words maximum.

I think sections 2 and 3 must be united in the section “Materials and Methods” as required by the journal format.

Figure 10 is unclear; focus on data (square zoom) does not refer to x-axis?

Round 2

Reviewer 3 Report

The authors have addressed all of my concerns.